

# Spin conductivity of the XXZ chain in the antiferromagnetic massive regime

**Frank Göhmann,[1] Karol K. Kozlowski,[2] Jesko Sirker,[3] Junji Suzuki[4]**

**1** Fakultät für Mathematik und Naturwissenschaften, Bergische Universität Wuppertal,
42097 Wuppertal, Germany
**2** Univ Lyon, ENS de Lyon, Univ Claude Bernard, CNRS, Laboratoire de Physique,
F-69342 Lyon, France
**3** Department of Physics and Astronomy and Manitoba Quantum Institute,
University of Manitoba, Winnipeg R3T 2N2, Canada
**4** Department of Physics, Faculty of Science, Shizuoka University,
Ohya 836, Suruga, Shizuoka, Japan

## Abstract

We present a series representation for the dynamical two-point function of the local spin current for the XXZ chain in the antiferromagnetic massive regime at zero temperature. From this series we can compute the correlation function with very high accuracy up to very long times and large distances. Each term in the series corresponds to the contribution of all scattering states of an even number of excitations. These excitations can be interpreted in terms of an equal number of particles and holes. The lowest term in the series comprises all scattering states of one hole and one particle. This term determines the long-time large-distance asymptotic behaviour which can be obtained explicitly from a saddle-point analysis. The space-time Fourier transform of the two-point function of currents at zero momentum gives the optical spin conductivity of the model. We obtain highly accurate numerical estimates for this quantity by numerically Fourier transforming our data. For the one-particle, one-hole contribution, equivalently interpreted as a two-spinon contribution, we obtain an exact and explicit expression in terms of known special functions. For large enough anisotropy, the two-spinon contribution carries most of the spectral weight, as can be seen by calculating the f-sum rule.



# 1 Introduction

Transport phenomena in spatially one-dimensional quantum systems are an active area of research, both theoretically and experimentally [1–10]. Some of the more prominent one-dimensional models are integrable and therefore amenable to an exact treatment. In linear response, their transport properties are determined by the dynamical correlation functions of current densities. In this work we shall focus on the XXZ spin-1/2 chain with Hamiltonian

$$H = J \sum_{j=1}^{L} \left\{ \sigma_{j-1}^x \sigma_j^x + \sigma_{j-1}^y \sigma_j^y + \Delta \left( \sigma_{j-1}^z \sigma_j^z - 1 \right) \right\} - \frac{h}{2} \sum_{j=1}^{L} \sigma_j^z, \tag{1}$$

where the $\sigma^\alpha \in \operatorname{End} \mathbb{C}^2$, $\alpha = x, y, z$, are Pauli matrices. The three real parameters of the Hamiltonian are the anisotropy $\Delta$, the exchange interaction $J > 0$, and the strength $h > 0$ of an external longitudinal magnetic field.

The basic quantities that can be transported in the XXZ chain are heat and magnetization. The total heat current of the XXZ chain is a conserved quantity. This implies that the corresponding thermal conductivity is purely ballistic and is determined entirely by a thermal Drude weight that can be calculated exactly at any temperature for any value of $\Delta$ and $h$ [11–13]. Technically, the thermal Drude weight can be inferred from the spectral properties of a properly defined quantum transfer matrix [11, 14–16].

For the current of the magnetization the situation is different. The total spin current is not conserved, except in the free Fermion case $\Delta = 0$. Still, it may have a ballistic contribution. In that case the corresponding conductivity consists of a singular 'dc part' quantified by a Drude weight and a regular $\omega$-dependent 'ac part', where $\omega$ is the frequency. There is a vast body of literature on the numerical calculation of both, the singular and the regular contribution, at finite temperature $T > 0$ (for an overview see the recent review article [1]). On the analytical side, the $T = 0$ Drude weight in the critical regime of the XXZ chain is known [17].

Results for the Drude weight at finite $T$ and on the leading asymptotic behaviour of the regular part for $\omega \to 0$ are also available. In particular, exact lower bounds, based on the Mazur inequality [18], were established in both cases [3, 19–24]. For the Drude weight in the regime $-1 < \Delta < 1$ at magnetic field $h = 0$ it has been argued that the Mazur bound obtained by taking all known families of conserved charges into account is tight. This bound, furthermore, does agree with earlier results for the Drude weight based on an extension of the thermodynamic Bethe ansatz [25–27]. In the latter case, the input invoked from the Bethe Ansatz solution of the XXZ chain [28, 29] enters in the form of the string hypothesis [30, 31], whose applicability is not established beyond the calculation of thermodynamic quantities, where its use is equivalent to the use of fusion hierarchies [32, 33]. For the low-energy excitations of the XXZ chain over the degenerate ground state in the antiferromagnetic massive regime of the phase diagram, considered in this work, it is known to give an incorrect description [29, 34, 35].

It is important to stress that all these works deal with the conductivity of the XXZ chain in the limit $\omega \to 0$; we are not aware of any exact result for genuine finite frequencies in the literature. In any case, our work presented below is independent of and rather orthogonal to the previous results in that, instead of $T > 0$, $\omega = 0$, we consider $T = 0$, $\omega > 0$ in the framework of an exact calculation of a dynamical correlation function that does not involve any kind of string hypothesis.

So far, the most successful attempts to exactly calculate dynamical correlation functions of the XXZ chain were based on different types of form factor series expansions. Besides the series involving form factors of the Hamiltonian [36–45], there is a different type of series that utilizes form factors of the quantum transfer matrix of the model [46]. The latter type has been dubbed the thermal form factor expansion. It was designed to deal with the canonical finite temperature case, but, in principle, can be extended to include certain generalized Gibbs ensembles [47]. The thermal form factor expansion has not been much explored so far, but it seems to have certain advantages over the more conventional expansions employing the form factors in a Hamiltonian basis. In [48, 49] it was observed that it can have rather nice asymptotic properties as compared to the conventional representation [50, 51] and can be interpreted as a resummation of the latter. For the XXZ chain in the massive antiferromagnetic regime, we observed a remarkable simplification of the Bethe root patterns occurring in the low-temperature limit [52]. In this limit all string excitations disappear and the whole spectrum of the quantum transfer matrix can be interpreted in terms of particle-hole excitations. This made it possible to derive a series representation for the longitudinal correlation functions of the XXZ chain in the massive antiferromagnetic regime, in which the $n$th term comprises all scattering states of $n$ particles and $n$ holes in a $2n$-fold integral with an integrand that is explicitly expressed in terms of known special functions [53, 54]. This has to be contrasted with older representations in which the integrand for general $n$ is itself a sum over multiple integrals [36], a multiple residue [35] or, in the best case, a product of explicit functions and Fredholm determinants [55]. It is the simple explicit form in [53, 54] which makes the higher-$n$ terms in the form factor series efficiently computable. The observed fast convergence of the series, on the other hand, is not a feature of the thermal form factor approach, but can be attributed to the massive nature of the excitations.

In Ref. [46], thermal form factor series were introduced with the example of operators of 'length one', where the length of an operator is defined as the number of lattice sites on which it acts non-trivially. The local Pauli matrices $\sigma_j^\alpha$ are examples of such operators. The local spin current

$$\jmath_j = -2\mathrm{i}J(\sigma_{j-1}^- \sigma_j^+ - \sigma_{j-1}^+ \sigma_j^-) \tag{2}$$

has length two. The results of Ref. [46] can be rather naturally extended to operators of arbitrary length. Although the general formula is easy to guess, its proof is slightly technical. It will be presented in a forthcoming publication. For a subclass of these operators (the

'spin-zero' operators) we can then introduce certain 'properly normalized form factors' which can be related to the theory of factorizing correlation functions and the Fermionic basis approach [56–60]. This allows to obtain series representations which, in the antiferromagnetic massive regime, are as explicit as the series for the longitudinal correlation functions obtained in [53, 54]. Prior to working out the general theory, we shall present here our results for the current-current correlation functions $\langle \mathcal{J}_1(t)\mathcal{J}_{m+1}\rangle$ that may be of particular interest to the physics community. For this special case the result may be obtained with moderate effort by combining [57, 60] with [46, 53, 54]. One has to start with a two-site generalized density matrix involving a twist or 'disorder parameter' $\alpha$ and then use the $R$-matrix symmetry, that imposes a set of quadratic relations on the two-site generalized density matrix, together with the reduction relation for the latter. The general case is harder and requires the techniques introduced in [58, 59].

## 2 Two-point function of currents

### 2.1 Form factor series

The antiferromagnetic massive regime of the ground state phase diagram of Hamiltonian (1) is defined by the inequalities $\Delta = \mathrm{ch}(\gamma) > 1$ and $|h| < h_\ell = 4J\,\mathrm{sh}(\gamma)\vartheta_4^2(0|q)$ for $\gamma > 0$. Here we have set $q = \mathrm{e}^{-\gamma}$, and $\vartheta_4$ denotes a Jacobi theta function. For the Jacobi theta functions, that will be frequently needed below, we shall follow the conventions of Ref. [61], see (37)-(38) for a reminder. Other special functions that occur in the definition of the form factor amplitudes belong to the families of $q$-gamma and $q$-hypergeometric (or basic hypergeometric) functions. We list their definitions and some of their properties in Appendix A.

In order to be able to present our series representation for the current-current correlation functions, we first of all have to fix some notation. In the antiferromagnetic massive regime, the dispersion relation of the elementary excitations can be expressed explicitly in terms of theta functions

$$p(\lambda) = \frac{\pi}{2} + \lambda - i\ln\left(\frac{\vartheta_4(\lambda + i\gamma/2|q^2)}{\vartheta_4(\lambda - i\gamma/2|q^2)}\right), \tag{3a}$$

$$\varepsilon(\lambda) = -2J\,\mathrm{sh}(\gamma)\vartheta_3\vartheta_4\frac{\vartheta_3(\lambda)}{\vartheta_4(\lambda)}. \tag{3b}$$

Here $p$ is the momentum, $\varepsilon$ is the dressed energy (for $h = 0$), $\lambda$ the rapidity, and we will use the convention $\vartheta_j = \vartheta_j(0|q)$.

The integrands in each term of our form factor series are parameterized in terms of two sets $\mathcal{U} = \{u_j\}_{j=1}^\ell$ and $\mathcal{V} = \{v_k\}_{k=1}^\ell$ of 'hole and particle type' rapidity variables of equal cardinality $\ell$. For sums and products over these variables we shall employ the short-hand notations

$$\sum_{\lambda\in\mathcal{U}\ominus\mathcal{V}} f(\lambda) = \sum_{\lambda\in\mathcal{U}} f(\lambda) - \sum_{\lambda\in\mathcal{V}} f(\lambda), \quad \prod_{\lambda\in\mathcal{U}\ominus\mathcal{V}} f(\lambda) = \frac{\prod_{\lambda\in\mathcal{U}} f(\lambda)}{\prod_{\lambda\in\mathcal{V}} f(\lambda)}. \tag{4}$$

We define

$$\Sigma = -\frac{\pi k}{2} - \frac{1}{2}\sum_{\lambda\in\mathcal{U}\ominus\mathcal{V}} \lambda \tag{5}$$

and

$$\phi^{(\pm)}(\lambda) = \mathrm{e}^{\pm i\Sigma}\prod_{\mu\in\mathcal{U}\ominus\mathcal{V}} \Gamma_{q^4}\left(\tfrac{1}{2} \pm \tfrac{\lambda-\mu}{2i\gamma}\right)\Gamma_{q^4}\left(1 \mp \tfrac{\lambda-\mu}{2i\gamma}\right). \tag{6}$$

We introduce multiplicative spectral parameters $H_j = \mathrm{e}^{2\mathrm{i}u_j}$, $P_k = \mathrm{e}^{2\mathrm{i}v_k}$ and the following special basic hypergeometric series,

$$\Phi_1(P_k, \alpha) = {}_{2\ell}\Phi_{2\ell-1}\begin{pmatrix} q^{-2}, \{q^2 \tfrac{P_k}{P_m}\}_{m\neq k}^\ell, \{\tfrac{P_k}{H_m}\}_{m=1}^\ell \\ \{\tfrac{P_k}{P_m}\}_{m\neq k}^\ell, \{q^2 \tfrac{P_k}{H_m}\}_{m=1}^\ell \end{pmatrix} ; q^4, q^{4+2\alpha} \Big), \tag{7a}$$

$$\Phi_2(P_k, P_j, \alpha) = {}_{2\ell}\Phi_{2\ell-1}\begin{pmatrix} q^6, q^2 \tfrac{P_j}{P_k}, \{q^6 \tfrac{P_j}{P_m}\}_{m\neq k,j}^\ell, \{q^4 \tfrac{P_j}{H_m}\}_{m=1}^\ell \\ q^8 \tfrac{P_j}{P_k}, \{q^4 \tfrac{P_j}{P_m}\}_{m\neq k,j}^\ell, \{q^6 \tfrac{P_j}{H_m}\}_{m=1}^\ell \end{pmatrix} ; q^4, q^{4+2\alpha} \Big). \tag{7b}$$

We further define

$$\Psi_2(P_k, P_j, \alpha) = q^{2\alpha} r_\ell(P_k, P_j)\Phi_2(P_k, P_j, \alpha), \tag{8}$$

where

$$r_\ell(P_k, P_j) = \frac{q^2(1-q^2)^2 \tfrac{P_j}{P_k}}{(1-\tfrac{P_j}{P_k})(1-q^4\tfrac{P_j}{P_k})}\Bigg[ \prod_{\substack{m=1 \\ m\neq j,k}}^\ell \frac{1-q^2\tfrac{P_j}{P_m}}{1-\tfrac{P_j}{P_m}} \Bigg]\Bigg[ \prod_{m=1}^\ell \frac{1-\tfrac{P_j}{H_m}}{1-q^2\tfrac{P_j}{H_m}} \Bigg], \tag{9}$$

and introduce a 'conjugation' $\overline{f}(H_j, P_k, q^\alpha) = f(1/H_j, 1/P_k, q^{-\alpha})$.

This allows to define the core part of our form factor densities, which is a matrix $\mathcal{M}$ with matrix elements

$$\mathcal{M}_{i,j} = \delta_{ij}\Big[ \overline{\Phi}_1(P_j, 0) - \frac{\phi^{(-)}(v_j)}{\phi^{(+)}(v_j)}\Phi_1(P_j, 0) \Big] - (1-\delta_{ij})\Big[ \overline{\Psi}_2(P_j, P_i, 0) - \frac{\phi^{(-)}(v_i)}{\phi^{(+)}(v_i)}\Psi_2(P_j, P_i, 0) \Big]. \tag{10}$$

By $\hat{\mathcal{M}}$ we denote the matrix obtained from $\mathcal{M}$ upon replacing $u_j \leftrightarrows -v_j$. We finally introduce one more function

$$\Xi(\lambda) = \frac{\Gamma_{q^4}\big(\tfrac{1}{2} + \tfrac{\lambda}{2\mathrm{i}\gamma}\big)G_{q^4}^2\big(1 + \tfrac{\lambda}{2\mathrm{i}\gamma}\big)}{\Gamma_{q^4}\big(1 + \tfrac{\lambda}{2\mathrm{i}\gamma}\big)G_{q^4}^2\big(\tfrac{1}{2} + \tfrac{\lambda}{2\mathrm{i}\gamma}\big)}. \tag{11}$$

Then the form factor amplitudes of the current-current correlation functions are

$$\mathcal{A}_{\mathcal{J}}^{(2\ell)}(\mathcal{U}, \mathcal{V}|k) = \Big( \frac{\sum_{\lambda \in \mathcal{U} \ominus \mathcal{V}} \varepsilon(\lambda)}{4\vartheta_1(\Sigma)/\vartheta_1'} \Big)^2 \Big[ \prod_{\lambda,\mu \in \mathcal{U} \ominus \mathcal{V}} \Xi(\lambda - \mu) \Big]\det_\ell\{\mathcal{M}\}\det_\ell\{\hat{\mathcal{M}}\}\det_\ell\Big( \frac{1}{\sin(u_j - v_k)} \Big)^2, \tag{12}$$

where $\varepsilon(\lambda)$ is the dressed energy and $\vartheta_1' = \vartheta_1'(0|q)$.

Using these amplitudes as well as the momentum and dressed energy defined in (3) we can formulate our main result. For every $m \geq 0$, the dynamical two-point function of spin currents of the XXZ chain in the antiferromagnetic massive regime and in the zero-temperature limit can be represented by the form-factor series

$$\langle \mathcal{J}_1(t)\mathcal{J}_{m+1} \rangle = \sum_{\ell=1}^\infty C^{(2\ell)}(m, t), \tag{13}$$

where

$$C^{(2\ell)}(m, t) = \sum_{k=0,1} \frac{(-1)^{mk}}{(\ell!)^2} \int_{\mathcal{C}_h^\ell} \frac{\mathrm{d}^\ell u}{(2\pi)^\ell} \int_{\mathcal{C}_p^\ell} \frac{\mathrm{d}^\ell v}{(2\pi)^\ell} \mathcal{A}_{\mathcal{J}}^{(2\ell)}(\mathcal{U}, \mathcal{V}|k) \mathrm{e}^{\mathrm{i}\sum_{\lambda \in \mathcal{U} \ominus \mathcal{V}}(t\varepsilon(\lambda) - mp(\lambda))} \tag{14}$$

is the $\ell$-particle $\ell$-hole contribution. The integration contours in (14) can be chosen as $\mathcal{C}_h = [-\tfrac{\pi}{2}, \tfrac{\pi}{2}] - \tfrac{\mathrm{i}\gamma}{2} + \mathrm{i}\delta$ and $\mathcal{C}_p = [-\tfrac{\pi}{2}, \tfrac{\pi}{2}] + \tfrac{\mathrm{i}\gamma}{2} + \mathrm{i}\delta'$, where $\delta, \delta' > 0$ are small. The derivation of (13), (14) is slightly cumbersome. It relies on a generalization of our work in Ref. [46] that will be published elsewhere and on the technical achievements obtained in Refs. [53,54].

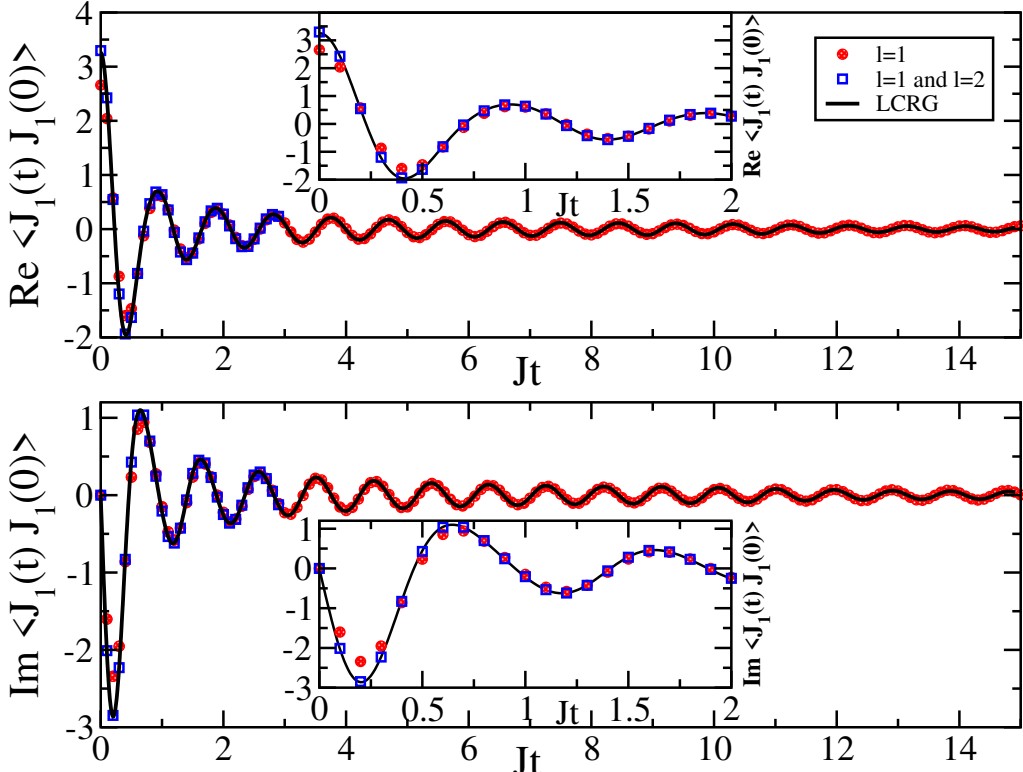

Figure 1: The real and imaginary contributions of the $\ell = 1$ term and of the sum of $\ell = 1$ and $\ell = 2$ terms to $\langle \mathcal{J}_1(t)\mathcal{J}_1 \rangle$ for $\Delta = 1.2$ are compared to LCRG results. The insets show that the $\ell = 2$ contribution becomes negligible on this scale for $tJ > 2$.

## 2.2 Spin-current correlations

Except for the vicinity of the isotropic point, the contributions from higher $\ell$ terms in (13) to $\langle \mathcal{J}_1(t)\mathcal{J}_{m+1} \rangle$ turn out to be small and to decrease rapidly in time. We plot the contributions of the $\ell = 1$ term and of the sum of the $\ell = 1$ and $\ell = 2$ terms to the current autocorrelation function ($m = 0$) for $\Delta = 1.2$ in Fig. 1.

In order to estimate the truncation error of our exact series representation we compare with independent exact results for $t = 0$. For small $m$ such results are available due to the factorization of the reduced density matrix in the static case [57–59]. We have checked that, for $0 \leq m \leq 2$ with various values of $\Delta$, away from the isotropic point, the sum of the $\ell = 1$ and $\ell = 2$ terms in (13) recovers the known exact values with good accuracy. For example, for $\Delta = 1.5$, the exact value of $\langle \mathcal{J}_1(0)\mathcal{J}_2 \rangle$ is $-0.333748\ldots$, while (13) yields $-0.333687\ldots$.

For $t > 0$ independent exact data are no longer available and the results of the form factor series are compared to a light-cone renormalization group (LCRG) calculation. The latter is a density-matrix renormalization group algorithm which makes use of the Lieb-Robinson bounds to obtain results for infinite chain lengths [62, 63]. In the LCRG calculations, we keep 8192 states in the truncated Hilbert space. The truncation error reaches $\sim 10^{-6}$ at the longest simulation times shown. A comparison with calculations where the number of states kept is varied (not shown) suggests that the error of the LCRG calculations remains always smaller than the size of the symbols used to represent the results from the form factor series. The LCRG data and the form factor series are in excellent agreement.

The explicit formula (13) makes it possible to evaluate the correlation function $\langle \mathcal{J}_1(t)\mathcal{J} \rangle \equiv \sum_{m=0}^{L_c} \langle \mathcal{J}_1(t)\mathcal{J}_{m+1} \rangle$, where $L_c \in \mathbb{N}$, even for large $L_c$ and $t$ numerically. As an illustration, we plot the real and imaginary parts of $\langle \mathcal{J}_1(t)\mathcal{J} \rangle$ for $\Delta = 3$ with $L_c = 349$ in Fig-

ure 2. The contributions from higher $\ell$ are small in this case and we only include $\ell = 1$. The high-frequency oscillation in the right plot, centred around zero, and the continuous decay of the correlation function for long times clearly indicate that, in agreement with common belief, the zero-temperature spin transport is non-ballistic. A ballistic contribution would show up as a non-vanishing constant long-time asymptotics of this correlation function.

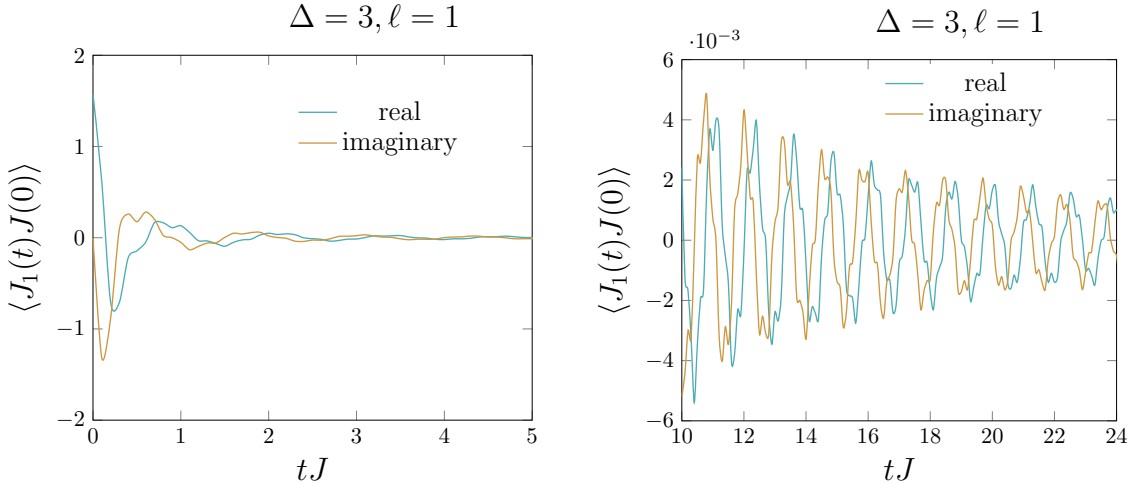

Figure 2: $\langle \mathcal{J}_1(t)\mathcal{J} \rangle$ for $\Delta = 3$, $L_c = 349$ and times $0 < tJ < 5$ (left), $10 < tJ < 24$ (right).

For smaller $\Delta$, the conclusion is less obvious from our data for small times, see the left panel in Fig. 3. However, in contrast to purely numerical methods the thermal form factor expansion allows to obtain highly accurate data for long times, see the right panel in Fig. 3. These data clearly show that the correlation function decays and has low-frequency oscillations.

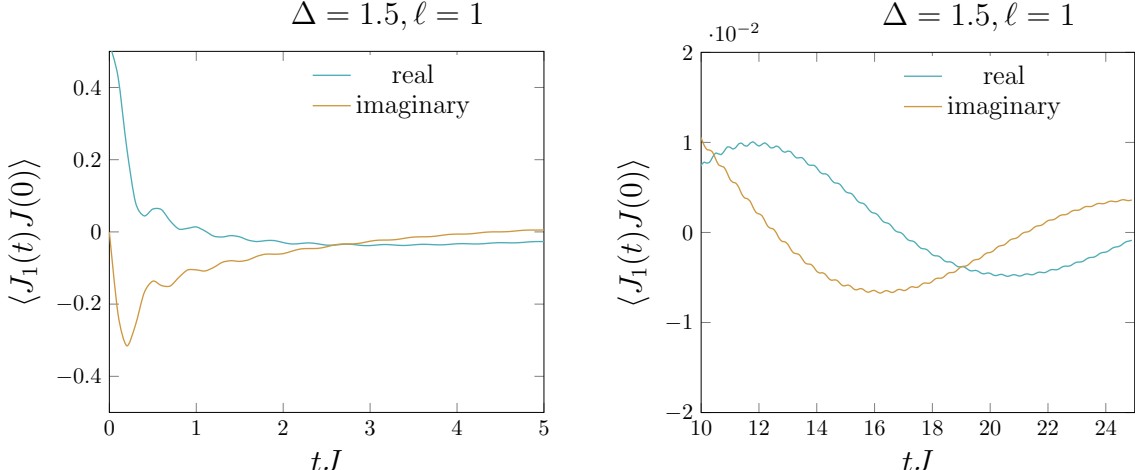

Figure 3: $\langle \mathcal{J}_1(t)\mathcal{J} \rangle$ for $\Delta = 1.5$ and times $0 < tJ < 5$ (left), $10 < tJ < 24$ (right). We used $L_c = 219$.

The upper limit $L_c$ in the sum over distances between the current operators is determined by a characteristic velocity of the excitations and the maximum time scale we want to reach. We fix $L_c \gtrsim v_2 t$, where $v_2$ is the upper critical velocity appearing in the long-time large-distance asymptotic analysis of the current-current correlation functions (see Sec. 2.3 below). Typical values of $v_2$ are listed in Table 1. We will choose similar values of $L_c$ in the sections below unless the contributions from large distances $m$ are negligible.

Table 1: Velocity $v_2/J$ for various anisotropies $\Delta$ according to equation (19).

| $\Delta$ | 1.5 | 2 | 3 |
|---|---|---|---|
| $v_2/J$ | 7.47329 | 9.06159 | 12.6851 |

## 2.3 Long-time large-distance asymptotics

From the example in Fig. 1, we see that the two-particle two-hole term significantly contributes to the numerical value of the correlation function only at short times. The long-time large-distance asymptotics of the correlation function is entirely determined by the one-particle one-hole term,

$$\left\langle \mathcal{J}_1(t)\mathcal{J}_{m+1} \right\rangle \sim C^{(2)}(m,t). \tag{15}$$

This is what makes the series representation (13), (14) so efficient. The double integral $C^{(2)}(m,t)$ can be numerically evaluated as accurately as we wish, because its asymptotic behaviour for $m, t \to \infty$ at fixed ratio $v = m/t$ is known in closed form from a saddle-point analysis. Such type of analysis was carried out for the two-point functions of the local magnetization $\langle \sigma_1^z(t)\sigma_{m+1}^z \rangle$ in one of our previous works [64]. Here the mathematical problem is exactly the same. Referring to equations (78), (79) in Appendix C, we can rewrite $C^{(2)}(m,t)$ as

$$C^{(2)}(m,t) = \int_{-\frac{\pi}{2}}^{\frac{\pi}{2}} \frac{\mathrm{d}z_1}{2\pi} \int_{-\frac{\pi}{2}}^{\frac{\pi}{2}} \frac{\mathrm{d}z_2}{2\pi} \, f(z_1,z_2) \, \mathrm{e}^{t(g(z_1)+g(z_2))}, \tag{16}$$

where

$$g(z) = \mathrm{i}\big(\varepsilon(z) - v p(z)\big), \tag{17a}$$

$$f(z_1,z_2) = \mathcal{A}_s^{(2)}(z_1,z_2|0) + (-1)^m \mathcal{A}_s^{(2)}(z_1,z_2|1). \tag{17b}$$

The definition of $\mathcal{A}_s^{(2)}$ can be found in equation (79) below. The important point is that $f(z_1,z_2)$ has a double zero at $z_1 = z_2$ implying that (16) is of the same form as the integral analysed in [64].

Let us briefly recall the main results of [64]. The asymptotics of $C^{(2)}$ is determined by the roots of the saddle-point equation $g'(z) = 0$ on steepest descent contours joining $-\pi/2$ and $\pi/2$. The saddle-point equation is most compactly expressed in terms of Jacobi elliptic functions and their parameters that will also occur below in our discussion of the one-particle one-hole contribution to the optical conductivity. We shall need the elliptic module $k$, the complementary module $k'$ and the complete elliptic integral $K$. They are all conveniently parameterized in terms of the elliptic nome $q$ by means of $\vartheta_j \equiv \vartheta_j(0|q)$,

$$k = \vartheta_2^2/\vartheta_3^2, \quad k' = \vartheta_4^2/\vartheta_3^2, \quad K = \pi\vartheta_3^2/2. \tag{18}$$

Let

$$v_1 = \frac{4JKk^2\,\mathrm{sh}(\gamma)}{\pi(1+k')}, \quad v_2 = \frac{4JKk^2\,\mathrm{sh}(\gamma)}{\pi(1-k')} \tag{19}$$

and $k_1 = v_1/v_2$. The first relation (18) is invertible which allows us to interpret $K$ as a function of $k$. Let $K_1 = K(k_1)$. Then the saddle-point equation can be represented as

$$\mathrm{sn}\left(\frac{4K_1 z}{\pi}\bigg|k_1\right) = \frac{v}{v_1}, \tag{20}$$

where sn is a Jacobi elliptic function. The solutions of the saddle-point equation divide the $m$-$t$ world plane into three different asymptotic regimes,

$$0 < v < v_1, \quad v_1 < v < v_2, \quad v_2 < v, \tag{21}$$

which were called [64] the 'time-like regime', the 'precursor regime' and the 'space-like regime' by analogy with the asymptotic analysis of electro-magnetic wave propagation in media.

Here we recall only the result of the asymptotic analysis in the time-like regime as it is relevant for the 'true long-time behaviour'. For the other two asymptotic regimes the reader is referred to [64]. In the time-like regime, $0 < v/v_1 < 1$, the saddle-point equation (20) has two inequivalent real solutions,

$$\lambda^- = \frac{\pi}{4K_1} \operatorname{arcsn}\left(\frac{v}{v_1}\bigg| k_1\right), \quad \lambda^+ = \frac{\pi}{2} - \lambda^-, \tag{22}$$

located in the interval $[0, \pi/2]$. The function occurring on the right hand side is the inverse Jacobi-sn function. The long-time large-distance asymptotics of $C^{(2)}(m, t)$ in the time-like regime is then determined by the saddle points,

$$C^{(2)}(m, t) \sim \frac{f(\lambda^+, \lambda^-)}{\pi t} \prod_{\sigma = \pm} \frac{e^{tg(\lambda^\sigma)}}{\sqrt{g''(\lambda^\sigma)}}. \tag{23}$$

Note that the product on the right hand side can be expressed explicitly in terms of elementary transcendental functions of $v = m/t$ [64].

# 3 Optical Conductivity

Quite generally, current-current correlation functions determine transport coefficients within the framework of linear response theory. The correlation function of two spin-current operators considered above determines the optical spin-conductivity $\sigma(\omega)$.

## 3.1 Form factor series

Several equivalent formula expressing $\sigma(\omega)$ for the XXZ chain in terms of current-current correlation functions have been described in the literature (see e.g. [1,4]). For our convenience and in order to make this work more self-contained, we have included concise derivations in Appendix B. We are interested in the thermodynamic limit in which the following lemma holds true.

**Lemma 1.** *The real part of the optical spin conductivity of the XXZ chain can be represented as*

$$\operatorname{Re}\sigma(\omega) = \frac{1 - e^{-\frac{\omega}{T}}}{2\omega} \int_{-\infty}^{\infty} dt \ e^{i\omega t} \left(2 \sum_{m=0}^{\infty} \langle \mathcal{J}_1(t)\mathcal{J}_{m+1}\rangle_T - \langle \mathcal{J}_1(t)\mathcal{J}_1\rangle_T\right). \tag{24}$$

*Proof.* We start with equation (70) from Appendix B,

$$\operatorname{Re}\sigma(\omega) = \frac{1 - e^{-\frac{\omega}{T}}}{2\omega} \int_{-\infty}^{\infty} dt \ e^{i\omega t} \lim_{L\to\infty} \frac{1}{L} \sum_{j,m=1}^{L} \langle \mathcal{J}_j(t)\mathcal{J}_m\rangle_T. \tag{25}$$

Here $L$ is the length of the periodic system. We shall assume $L$ to be even. The Hamiltonian is invariant under translations modulo $L$ and under parity transformations $j \to L - j + 1$. Hence,

$$\frac{1}{L}\sum_{j,m=1}^{L}\left\langle \mathcal{J}_j(t)\mathcal{J}_m\right\rangle_T = \sum_{m=1}^{L}\left\langle \mathcal{J}_1(t)\mathcal{J}_m\right\rangle_T$$

$$= \left\langle \mathcal{J}_1(t)\mathcal{J}_1\right\rangle_T + \sum_{m=1}^{\frac{L}{2}-1}\left\langle \mathcal{J}_1(t)\mathcal{J}_{m+1}\right\rangle_T + \left\langle \mathcal{J}_1(t)\mathcal{J}_{\frac{L}{2}+1}\right\rangle_T + \sum_{m=\frac{L}{2}+1}^{L-1}\left\langle \mathcal{J}_1(t)\mathcal{J}_{m+1}\right\rangle_T$$

$$= \left\langle \mathcal{J}_1(t)\mathcal{J}_1\right\rangle_T + 2\sum_{m=1}^{\frac{L}{2}-1}\left\langle \mathcal{J}_1(t)\mathcal{J}_{m+1}\right\rangle_T + \left\langle \mathcal{J}_1(t)\mathcal{J}_{\frac{L}{2}+1}\right\rangle_T. \quad (26)$$

Here, we have split the summation so that, upon using the $L$-periodicity of the lattice, the summed-up terms do get farther and farther away from the first site. This produces the factor of 2 and is necessary for appropriately taking the thermodynamic limit. Hence,

$$\lim_{L\to\infty}\frac{1}{L}\sum_{j,m=1}^{L}\left\langle \mathcal{J}_j(t)\mathcal{J}_m\right\rangle_T = \left\langle \mathcal{J}_1(t)\mathcal{J}_1\right\rangle_T + 2\sum_{m=1}^{\infty}\left\langle \mathcal{J}_1(t)\mathcal{J}_{m+1}\right\rangle_T, \quad (27)$$

where the expectation values on the right hand side are now those in the thermodynamic limit ($m$ fixed, $L \to \infty$). □

The zero-temperature limit of (24) for $\omega > 0$ is obvious. In this limit we can take our numerical results for $\langle \mathcal{J}_1(t)\mathcal{J}\rangle$ from section 2.2 with $L_c$ sufficiently large in order to compute $\mathrm{Re}\,\sigma(\omega)$ numerically from Lemma 1. The results shown in Fig. 4 and Fig. 6 below were obtained this way. On the other hand, the summation over all lattice sites involved in (24) can be easily carried out analytically on the series representation (13), (14), and we obtain the following lemma.

**Lemma 2.** *In the antiferromagnetic massive regime for $T \to 0$ the correlation function under the integral in (24) has the thermal form factor series representation*

$$2\sum_{m=0}^{\infty}\left\langle \mathcal{J}_1(t)\mathcal{J}_{m+1}\right\rangle - \left\langle \mathcal{J}_1(t)\mathcal{J}_1\right\rangle =$$

$$\sum_{\substack{\ell\in\mathbb{N}\\k=0,1}}\frac{1}{(\ell!)^2}\int_{\mathcal{C}_h^\ell}\frac{\mathrm{d}^\ell u}{(2\pi)^\ell}\int_{\mathcal{C}_p^\ell}\frac{\mathrm{d}^\ell v}{(2\pi)^\ell}\,\mathcal{A}_\sigma^{(2\ell)}(\mathcal{U},\mathcal{V}|k)\,\mathrm{e}^{\mathrm{i}t\sum_{\lambda\in\mathcal{U}\ominus\mathcal{V}}\varepsilon(\lambda|h)}, \quad (28)$$

*where*

$$\mathcal{A}_\sigma^{(2\ell)}(\mathcal{U},\mathcal{V}|k) = -\mathrm{i}\,\mathrm{ctg}\left(\tfrac{1}{2}\left(\pi k + \sum_{\lambda\in\mathcal{U}\ominus\mathcal{V}}p(\lambda)\right)\right)\mathcal{A}_{\mathcal{J}}^{(2\ell)}(\mathcal{U},\mathcal{V}|k). \quad (29)$$

*Proof.* $\mathrm{Re}\sum_{\lambda\in\mathcal{U}\ominus\mathcal{V}}\mathrm{i}p(\lambda) > 0$ if $u_j \in \mathcal{C}_h$ and $v_k \in \mathcal{C}_p$ as can, for instance, be seen from Appendix A.2 of [35]. Hence, the summation over $m$ can be performed by the geometric sum formula. □

With equation (28), we have an alternative starting point for a numerical computation of the real part of the optical conductivity at $T = 0$. Again, we would have to substitute this formula into (24) for $T = 0$, $\omega > 0$ and calculate the Fourier transform numerically. For the time being we refrain from this possibility. The direct use of (13) and (14), for which we could resort to existing computer programs [54], gives reliable numerical results as can be seen from Fig. 4. Our algorithm uses numerical saddle point integration. For a use with (28), (29) we would have to modify it as some of the poles of the integrand are now located at the saddle points. We leave the interesting questions of how to deal with this situation numerically and how to calculate the long-time asymptotics analytically for future studies.

### 3.2 Two-spinon contribution

In the general case of $\ell$ particle-hole excitations, the numerical evaluation of the Fourier transform seems to be the most efficient way to make use of Lemma 2. For the $\ell = 1$ particle-hole contribution, however, following the examples of [38, 42, 65], the Fourier transformation can be carried out by hand. The details of the calculation are discussed in Appendix C.

We introduce two functions

$$r(\omega) = \frac{\pi}{K} \operatorname{arcsn}\left( \frac{\sqrt{(h_\ell/k')^2 - \omega^2}}{h_\ell k/k'} \bigg| k \right), \tag{30a}$$

$$B(z) = \frac{1}{G_{q^4}^4\left(\frac{1}{2}\right)} \prod_{\sigma = \pm} \frac{G_{q^4}\left(1 + \frac{\sigma z}{2i\gamma}\right) G_{q^4}\left(\frac{\sigma z}{2i\gamma}\right)}{G_{q^4}\left(\frac{3}{2} + \frac{\sigma z}{2i\gamma}\right) G_{q^4}\left(\frac{1}{2} + \frac{\sigma z}{2i\gamma}\right)}, \tag{30b}$$

where $h_\ell$ has been defined at the beginning of Sec. 2.1. Using (18) and (30) we can formulate our result for the two-spinon contribution to the optical conductivity.

**Lemma 3.** *The two-spinon contribution to the real part of the optical conductivity of the XXZ chain at zero temperature and in the antiferromagnetic massive regime can be represented as*

$$\operatorname{Re}\sigma^{(2)}(\omega) = \frac{q^{\frac{1}{2}} h_\ell^2 k}{8k'} \frac{B(r(\omega))}{\Delta - \cos(r(\omega))} \frac{\vartheta_3^2}{\vartheta_3^2(r(\omega)/2)} \frac{1}{\sqrt{((h_\ell/k')^2 - \omega^2)(\omega^2 - h_\ell^2)}}, \tag{31}$$

*as long as $h_\ell < \omega < h_\ell/k'$. It vanishes outside of this range of $\omega$.*

The derivation of this result is discussed in Appendix C. In the above expression (31), we identify two van-Hove singularities at the upper and lower 2-spinon band edges, see the last factor on the right hand side. They are both canceled by $B(r(\omega))$ as $B(z)$ has double zeros at integer multiples of $\pi$. As a result, $\operatorname{Re}\sigma^{(2)}(\omega)$ has square-root singularities, both at the lower and the upper 2-spinon band edges, away from the isotropic point.

As a consistency test, we plot the spin conductivity obtained from a numerical Fourier transformation of the $\ell = 1$ parts of $\langle \mathcal{J}_1(t)\mathcal{J}_{m+1}\rangle$ and the above analytic result in Fig. 4. The two curves agree well except for very small frequencies $\omega$. Note that we perform the summation and the numerical integration in (25) without introducing any window functions or filters. The overall factor $\omega^{-1}$ in (25) could then introduce an artificial instability. We nevertheless observe only a small deviation from zero in the vicinity of $\omega = 0$. This indicates a high accuracy of our spin-current correlation data.

With decreasing anisotropy $\Delta$, the peak position moves towards $\omega = 0$ while its height increases, see Fig. 5. This is expected because the XXZ chain has a non-zero $T = 0$ Drude weight in the isotropic limit [17]. A short discussion of the isotropic limit is presented in Appendix C.

### 3.3 More than two spinons

For $\ell \geq 2$, simple analytic expressions like (31) are not available. Nevertheless, as has already been mentioned, the spin-current correlation function is enumerable for sufficiently large $m$ and $t$. This enables us to perform a direct Fourier transformation, at least in principle. In practice, calculating higher contributions is time-consuming. Here we therefore only present a result for $\ell = 2$, which corresponds to the 4-spinon case. Inside the two-spinon band the additional contribution from 4-spinon states is small, away from the isotropic point. Above the 2-spinon upper band edge, however, a small conductivity is entirely carried by the scattering states of four and more spinons. For $\Delta = 2, 3$ the maximum of the $\ell = 2$ contribution is located

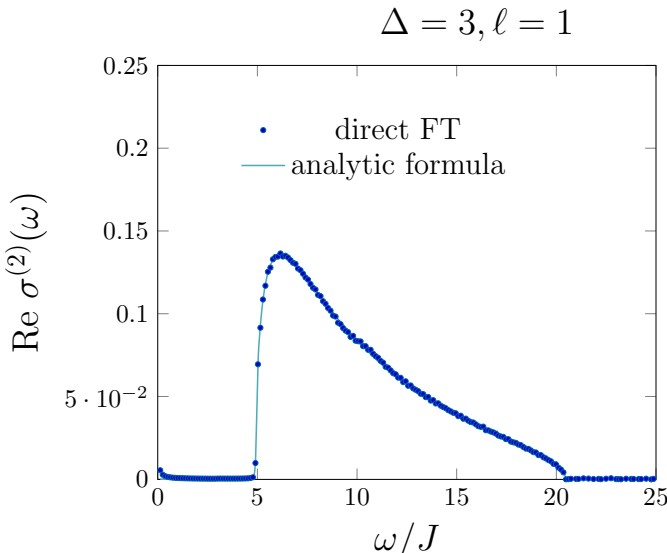

Figure 4: Comparison of the analytic result (31) and a numerical Fourier transformation of the $\ell = 1$ part of $\left\langle \mathcal{J}_1(t)\mathcal{J}_{m+1} \right\rangle$ for anisotropy $\Delta = 3$. For the latter we use $0 \le m \le 399$ and $0 \le tJ \le 50$.

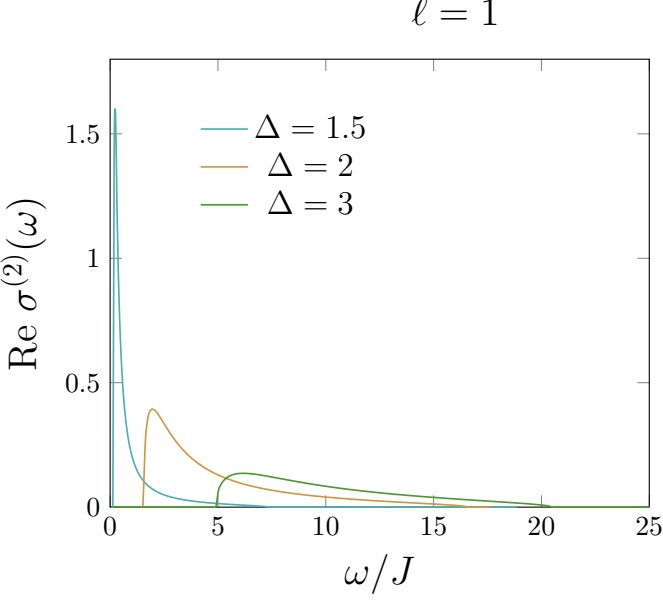

Figure 5: $\ell = 1$ contribution, Eq. (31), to $\operatorname{Re}\sigma^{(2)}(\omega)$ for various $\Delta$.

close to the upper 2-spinon band edge, and we expect this contribution to be significant even near the lower 2-spinon band edge as $\Delta \to 1$. An example for $\Delta = 3$ is shown in Fig. 6.

As a benchmark for the accuracy of our results we consider the $f$-sum rule [66],

$$\int_0^\infty \mathrm{d}\omega \, \operatorname{Re}\sigma(\omega) = -\lim_{L\to\infty} \frac{\pi\langle H_0\rangle}{2L}, \tag{32}$$

where $H_0$ denotes the 'kinetic part' of the Hamiltonian, obtained from (1) by setting $\Delta, h = 0$. The results of a numerical comparison of the left and right hand side of the sum rule (32) are summarized in table 2. We find, in particular, that for the chosen anisotropies the sum of the $\ell = 1$ and $\ell = 2$ terms is already extremely close to the full weight.

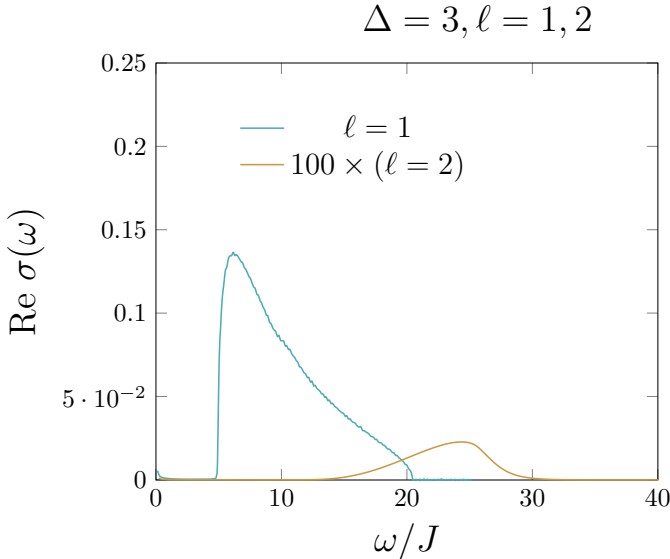

Figure 6: The $\ell = 1$ and $\ell = 2$ contributions to $\mathrm{Re}\,\sigma(\omega)$ for $\Delta = 3$. Note that the contribution from $\ell = 2$ is multiplied by a factor 100. For the Fourier transform for $\ell = 2$ we use data for $\langle \mathcal{J}_1(t)\mathcal{J}_{m+1}\rangle$ with $0 \le m \le 39$ and $0 \le tJ \le 30$.

Table 2: Both sides of the $f$-sum rule (32) for $\Delta = 1.5, 2$ and $3$ and $J = 1$.

| $\Delta$ | lhs of (32): $\ell = 1$ | lhs of (32): $(\ell = 1) + (\ell = 2)$ | rhs of (32) |
|---|---|---|---|
| 1.5 | 1.56692 | 1.64348 | 1.64394 |
| 2 | 1.36065 | 1.37615 | 1.37624 |
| 3 | 0.987313 | 0.989092 | 0.989116 |

## 4 Summary and Conclusions

We have presented an exact thermal form factor expansion for the dynamical current-current correlation function $\langle \mathcal{J}_1(t)\mathcal{J}_{m+1}\rangle$ of the spin-1/2 XXZ chain in the massive antiferromagnetic regime at zero temperature. In this expansion, the correlation function is given as a sum over $\ell$ particle-hole excitations or, equivalently, $2\ell$ spinon excitations. The formula can, in principle, be evaluated numerically for arbitrary distances $m$ and times $t$, leading to numerically exact results. We note, in particular, that the series in the number of particle-hole excitations $\ell$ converges fast, except for anisotropy $\Delta \to 1$. The long-time, large-distance asymptotics is determined by the $\ell = 1$ contribution. We attribute the fast convergence to the massive nature of the involved excitations.

We have also provided a form factor series representation for $\lim_{L\to\infty} \frac{1}{L}\sum_{j,m=1}^{L} \langle \mathcal{J}_j(t)\mathcal{J}_m\rangle$ which allows to calculate the real part of the optical spin conductivity $\mathrm{Re}\,\sigma(\omega)$ by a direct Fourier transform. For the $\ell = 1$ (2-spinon) contribution the Fourier transform can be performed analytically, leading to a closed form expression for the 2-spinon optical conductivity. We find that $\mathrm{Re}\,\sigma(\omega)$ is finite only within the 2-spinon band which starts at some finite frequency. At both edges of the spinon band, the conductivity shows a square-root behavior. By checking the $f$-sum rule, we have shown that the $\ell = 1$ and $\ell = 2$ contributions account for almost the entire spectral weight if we are not too close to the isotropic point.

Another test of our form factor series for $\left\langle \mathcal{J}_1(t)\mathcal{J}_{m+1}\right\rangle$ was provided by DMRG results. We also note that the obtained $\text{Re}\,\sigma(\omega)$ looks quite similar to the finite temperature results in Ref. [67], which were obtained by DMRG as well, except for small frequencies. The Lorentzian-type peak around $\omega = 0$ observed in this paper, which seems to decrease with increasing $T$, therefore appears to be a genuine finite-temperature effect related to the expected diffusive behavior. To understand the low-frequency behavior better, it would therefore be of interest to extend our form factor series expansion to finite temperatures.

This is not totally out of reach, since the thermal form factor approach is a genuine finite-temperature method which only has been used in the zero-temperature limit here to produce fully explicit results. One of our future goals is indeed to keep the temperature finite. For this purpose we will need better control of the non-linear integral equations that describe the excited states of the quantum transfer matrix. Simplifications should occur in the high-temperature limit, where we have a rather complete understanding [68] of the involved auxiliary functions.

Further future goals include a proof of the convergence of the series and an estimation of the truncation error. Given the explicit nature of the integrands in our series and the recent progress in related cases [49, 69] this may now appear within reach. We shall also work out thermal form factor series expansions of general two-point functions of spin zero operators and provide the details of the proof of (13), (14) in a forthcoming publication.

# Acknowledgments

The authors would like to thank Z. Bajnok, H. Boos, A. Klümper, F. Smirnov and A. Weiße for helpful discussions. FG and JSi acknowledge financial support by the German Research Council (DFG) in the framework of the research unit FOR 2316. KKK is supported by CNRS Grant PICS07877 and by the ERC Project LDRAM: ERC-2019-ADG Project 884584. JSi acknowledges support by the Natural Sciences and Engineering Research Council (NSERC, Canada). JSu is supported by a JSPS KAKENHI Grant number 18K03452.

# A  Special functions

In this appendix we gather the definitions of the special functions needed in the main part of the text and list some of their properties.

We start with functions that can be expressed in terms of infinite $q$-multi factorials which, for $|q_j| < 1$ and $a \in \mathbb{C}$, are defined as

$$(a; q_1, \ldots, q_p) = \prod_{n_1, \ldots, n_p = 0}^{\infty} (1 - a q_1^{n_1} \ldots q_p^{n_p}). \tag{33}$$

A first set of such functions are the $q$-Gamma and $q$-Barnes functions $\Gamma_q$ and $G_q$,

$$\Gamma_q(x) = (1-q)^{1-x} \frac{(q;q)}{(q^x;q)}, \tag{34a}$$

$$G_q(x) = (1-q)^{-\frac{1}{2}(1-x)(2-x)}(q;q)^{x-1} \frac{(q^x; q, q)}{(q; q, q)}. \tag{34b}$$

They satisfy the normalization conditions

$$\Gamma_q(1) = G_q(1) = 1 \tag{35}$$

and the basic functional equations

$$[x]_q \Gamma_q(x) = \Gamma_q(x+1), \quad \Gamma_q(x)G_q(x) = G_q(x+1), \tag{36}$$

where $[x]_q = (1-q^x)/(1-q)$ is a familiar form of the $q$-number.

Closely related are the Jacobi theta functions $\vartheta_j(x) = \vartheta_j(x|q)$, $j = 1, \ldots, 4$. Setting $q = \mathrm{e}^{-\gamma}$ they can be introduced by

$$\vartheta_4(x|q) = (q^2; q^2)(e^{-2\mathrm{i}x} q; q^2)(e^{2\mathrm{i}x} q; q^2) \tag{37}$$

and

$$\vartheta_1(x) = -\mathrm{i}q^{\frac{1}{4}} \mathrm{e}^{\mathrm{i}x} \vartheta_4(x + \mathrm{i}\gamma/2), \quad \vartheta_2(x) = q^{\frac{1}{4}} \mathrm{e}^{\mathrm{i}x} \vartheta_4(x + \mathrm{i}\gamma/2 + \pi/2),$$

$$\vartheta_3(x) = \vartheta_4(x + \pi/2). \tag{38}$$

The parameter $q$ of the theta functions is called 'the nome'. Sometimes we suppress their nome dependence, but only if the value of $q$ is clear from the context.

The Jacobi theta functions are connected with the $q$-gamma functions through the second functional relation of the latter,

$$\frac{\vartheta_4(x|q)}{\vartheta_4(0|q)} = \frac{\Gamma_{q^2}^2\left(\frac{1}{2}\right)}{\Gamma_{q^2}\left(\frac{1}{2} - \frac{\mathrm{i}x}{\gamma}\right)\Gamma_{q^2}\left(\frac{1}{2} + \frac{\mathrm{i}x}{\gamma}\right)}. \tag{39}$$

We shall also frequently employ the common notational convention for the 'theta constants', $\vartheta_j = \vartheta_j(0|q)$, $j = 2, 3, 4$, $\vartheta_1' = \vartheta_1'(0|q)$.

Another class of functions needed in the main text are the basic hypergeometric functions [70]. They are defined in terms of finite $q$ multi-factorials (or $q$-Pochhammer symbols),

$$(a_1, \ldots, a_k; q)_m = (a_1; q)_m (a_2; q)_m \ldots (a_k; q)_m, \quad (a; q)_m = \prod_{k=0}^{m-1} (1 - aq^k), \tag{40}$$

by the infinite series

$$_r\Phi_s\begin{pmatrix} a_1, \ldots, a_r \\ b_1, \ldots, b_s \end{pmatrix}; q, z = \sum_{k=0}^{\infty} \frac{(a_1, \ldots, a_r; q)_k}{(b_1, \ldots, b_s, q; q)} \left((-1)^k q^{\frac{k(k-1)}{2}}\right)^{s+1-r} z^k. \tag{41}$$

## B The spin conductivity of the XXZ chain

In this appendix we recall the derivation of several alternative formulae for the 'spin conductivity'.

### B.1 Gauge fields coupling to the Hamiltonian

We decompose the Hamiltonian (1) as

$$H = H_0 + \Delta H_I - hS^z, \tag{42}$$

where

$$H_0 = 2J \sum_{j=1}^{L} (\sigma_{j-1}^+ \sigma_j^- + \sigma_{j-1}^- \sigma_j^+), \quad H_I = J \sum_{j=1}^{L} (\sigma_{j-1}^z \sigma_j^z - 1). \tag{43}$$

Under a Jordan-Wigner transformation the operator $H_0$ goes to a tight-binding type Hamiltonian, while $H_I$ becomes a nearest-neighbour density-density interaction. In the Fermion picture, it is $H_0$ which couples to an external electro-magnetic field via so-called Peierls phases which can be understood as a manifestation of a $U(1)$ gauge field. For details see e.g. Chapter 1.3 of the book [71]. In the spin-chain picture, switching on an external field means to replace

$$\sigma_j^- \to e^{i\varphi_j(t)}\sigma_j^-, \quad \sigma_j^+ \to e^{-i\varphi_j(t)}\sigma_j^+, \tag{44}$$

where $t$ is the time variable. We shall restrict ourselves to a spatially homogeneous field ('the case of long wave length'),

$$\varphi_j(t) - \varphi_{j-1}(t) = \lambda(t). \tag{45}$$

Then $H_0$ turns into

$$H_\lambda = 2J \sum_{j=1}^{L} \left( e^{i\lambda(t)}\sigma_{j-1}^+ \sigma_j^- + e^{-i\lambda(t)}\sigma_{j-1}^- \sigma_j^+ \right). \tag{46}$$

By analogy with the electro-magnetic case we shall assume that the gauge field is related to the 'electric field' $E$ as

$$\partial_t \lambda(t) = -eaE(t), \tag{47}$$

where $e$ is a unit charge and $a$ a unit length ('lattice spacing'). This implies the relation

$$\lambda_F(\omega) = -\frac{iea}{\omega}E_F(\omega), \quad \text{with} \quad \lambda_F(\omega) = \int_{\mathbb{R}} dt\, e^{i\omega t}\lambda(t), \tag{48}$$

for the Fourier transforms. We shall consider a class of fields $\lambda$ for which $|\lambda(t)| \le e^{\varepsilon t}$ for $t \to -\infty$ and $|\lambda(t)| \le ct$ for $t \to \infty$, where $\varepsilon, c > 0$. The first condition is compatible with an adiabatic switching on of the field and the second one admits 'electric fields' which are asymptotically constant and allow us to probe the dc conductivity. For such fields the Fourier transform $\lambda_F(\omega)$ exists within a strip $0 < \text{Im}\,\omega < \varepsilon$ and should be interpreted as a '+-boundary value' on the real axis.

## B.2 Current operators

An external 'electric field' will induce a current into a wire. Let us recall the construction of the corresponding current operator.

We start with the definition of the operator of the time derivative of a physical quantity in the Schrödinger picture. The Schrödinger equation,

$$i\partial_t U(t) = H(t)U(t), \quad U(0) = \text{id}, \tag{49}$$

determines the time evolution operator $U(t)$ for a system with generally time dependent Hamiltonian $H(t)$. If $A$ is any operator in the Schrödinger picture, then the corresponding operator $A_H$ in the Heisenberg picture is

$$A_H(t) = U^{-1}(t)AU(t). \tag{50}$$

Equations (49) and (50) imply that

$$i\partial_t A_H(t) = -U^{-1}(t)[H(t),A]U(t) \tag{51}$$

or

$$U(t)(\partial_t A_H(t))U^{-1}(t) = i[H(t),A] = \dot{A}. \tag{52}$$

We may think of this equation as defining the time derivative $\dot{A}$ of $A$ in the Schrödinger picture.

Applying this to the local magnetization $\frac{1}{2}\sigma^z$ and $H(t) = H_\lambda + \Delta H_I - hS^z$ we obtain

$$\frac{1}{2}\dot{\sigma}_j^z = -J_{j+1}(t) + J_j(t), \tag{53}$$

where

$$J_j(t) = 2iJ\left(e^{i\lambda(t)}\sigma_{j-1}^+\sigma_j^- - e^{-i\lambda(t)}\sigma_{j-1}^-\sigma_j^+\right). \tag{54}$$

Equation (53) has the form of a continuity equation for the local magnetization. For this reason $J_j(t)$ is interpreted as the density of the spin current.

Let

$$\mathcal{J} = \sum_{m=1}^L J_m(0) = \sum_{m=1}^L \mathcal{J}_m. \tag{55}$$

Then the total magnetic current is the sum

$$J(t) = \sum_{m=1}^L J_m(t) = \mathcal{J} - \lambda(t)H_0 + \mathcal{O}(\lambda^2) \tag{56}$$

and the time dependent Hamiltonian has the expansion

$$H_\lambda + \Delta H_I - hS^z = H + \lambda(t)\mathcal{J} + \mathcal{O}(\lambda^2). \tag{57}$$

The latter two equations are all we need in order to calculate the average current induced by the external field within the linear response theory. The small time dependent perturbation we can read off from (57) is $V(t) = \lambda(t)\mathcal{J}$.

## B.3 Linear response of the current

We denote the density matrix of the canonical ensemble by $\rho_c$ and the density matrix obtained by time evolving $\rho_c$ with $H_\lambda$ by $\rho(t)$. Then the linear response formula

$$\text{tr}\left\{\left(\rho(t) - \rho_c\right)J(t)\right\} = -i\int_{-\infty}^t dt' \left\langle \left[(J(t))_H(t-t'), V(t')\right]\right\rangle_T \tag{58}$$

determines the averaged current to linear order in $V$ (for a concise derivation see e.g. Section L.22 of [72]). Inserting here (56) and (57) we obtain

$$\text{tr}\left\{\rho(t)J(t)\right\} = -\langle H_0\rangle_T \lambda(t) - i\int_{-\infty}^\infty dt'\, \Theta(t-t')\left\langle\left[\mathcal{J}(t-t'), \mathcal{J}\right]\right\rangle_T \lambda(t') + \mathcal{O}(\lambda^2). \tag{59}$$

In this equation $\Theta$ is the Heaviside step function and $\mathcal{J}(t)$ denotes the total current $\mathcal{J}$ in the Heisenberg picture that is evolved with respect to the unperturbed Hamiltonian $H$. We have made use of the fact that $\langle\mathcal{J}\rangle_T = 0$ due to the invariance of the XXZ Hamiltonian under parity transformations.

The 'experimentally relevant quantity' is the Fourier transformed current per volume which in physical units is given by

$$\mathcal{J}_F(\omega) = -ea\int_{-\infty}^\infty dt\, e^{i\omega t}\frac{\text{tr}\left\{\rho(t)J(t)\right\}}{a^3 L}. \tag{60}$$

Due to the remark below (48), the integral on the right hand side is to be interpreted as a $+$-boundary value if $\omega$ is real.

If we substitute (59) into (60), use the convolution theorem as well as (48) and neglect all terms of quadratic oder in $\lambda$ or higher, we arrive at 'Ohm's law',

$$\mathcal{J}_F(\omega) = \frac{e^2}{a}\sigma_L(\omega)E_F(\omega),\tag{61}$$

where $\sigma_L(\omega)$ is the specific optical conductivity,

$$\sigma_L(\omega) = \frac{1}{L(\omega + \mathrm{i}0)}\left\{-\mathrm{i}\langle H_0\rangle_T + \int_0^\infty \mathrm{d}t\ \mathrm{e}^{\mathrm{i}\omega t}\big\langle[\mathcal{J}(t),\mathcal{J}]\big\rangle_T\right\}.\tag{62}$$

Assuming $\big\langle[\mathcal{J}(t),\mathcal{J}]\big\rangle_T$ to be bounded for $t \to +\infty$ we see that the right hand side of (62) is a holomorphic function of $\omega$ in the upper half plane. This implies that the real part and the imaginary part of the optical conductivity are not independent, but are connected by the Kramers-Kronig relation.

## B.4  Real part of the optical conductivity

For this reason we can focus our attention on the real part of the conductivity. We wish to rewrite it in a form appropriate for taking the thermodynamic limit. We basically follow the arguments given in [4] and start by switching to a spectral representation of the integral on the right hand side of (62). Employing the notation

$$Z_\lambda = \mathrm{tr}\left\{\mathrm{e}^{-\frac{1}{T}(H_\lambda + \Delta H_I - hS^z)}\right\},\quad p_n = \frac{\mathrm{e}^{-\frac{E_n}{T}}}{Z_0},\quad \omega_{mn} = E_m - E_n,\tag{63}$$

where the $E_n$ are the eigenvalues of the Hamiltonian (1) with corresponding eigenstates $|n\rangle$, the spectral representation takes the form

$$\int_0^\infty \mathrm{d}t\ \mathrm{e}^{\mathrm{i}\omega t}\big\langle[\mathcal{J}(t),\mathcal{J}]\big\rangle_T = \mathrm{i}\sum_{\substack{m,n\\E_m\neq E_n}}\frac{p_n - p_m}{\omega - \omega_{mn} + \mathrm{i}0}\big|\langle m|\mathcal{J}|n\rangle\big|^2.\tag{64}$$

Now, if $E_m \neq E_n$,

$$\frac{1}{\omega + \mathrm{i}0}\cdot\frac{1}{\omega - \omega_{mn} + \mathrm{i}0} = \frac{1}{\omega_{mn}}\left(\frac{1}{\omega - \omega_{mn} + \mathrm{i}0} - \frac{1}{\omega + \mathrm{i}0}\right).\tag{65}$$

Using this identity as well as the Plemelj formula $1/(\omega + \mathrm{i}0) = -\mathrm{i}\pi\delta(\omega) + \mathcal{P}(1/\omega)$ we obtain the spectral representation

$$\mathrm{Re}\,\sigma_L(\omega) = \frac{\pi}{L}\left\{-\langle H_0\rangle_T + \sum_{\substack{m,n\\E_m\neq E_n}}\frac{p_m - p_n}{\omega_{mn}}\big|\langle m|\mathcal{J}|n\rangle\big|^2\right\}\delta(\omega)$$

$$-\frac{\pi}{L}\sum_{\substack{m,n\\E_m\neq E_n}}\frac{p_m - p_n}{\omega_{mn}}\big|\langle m|\mathcal{J}|n\rangle\big|^2\delta(\omega - \omega_{mn}).\tag{66}$$

This representation immediately implies the f-sum rule (32).

Now notice that the free energy per lattice site

$$f(\lambda) = -\frac{T}{L}\ln Z_\lambda\tag{67}$$

satisfies [73] the relation

$$\partial_\lambda^2 f(\lambda)\big|_{\lambda=0} = -\frac{\langle H_0 \rangle_T}{L} + \frac{1}{L} \sum_{\substack{m,n \\ E_m \neq E_n}} \frac{p_m - p_n}{\omega_{mn}} \big| \langle m | \jmath | n \rangle \big|^2 - \frac{1}{TL} \sum_{\substack{m,n \\ E_m \neq E_n}} p_m \big| \langle m | \jmath | n \rangle \big|^2. \tag{68}$$

This quantity is the so-called Meissner fraction. It vanishes in the thermodynamic limit [73], which follows from the fact that the effect of the external field $\lambda$ is equivalent to a mere twist of the boundary conditions of the original Hamiltonian (1). Inserting (68) into (66) and switching back from a spectral representation to a Fourier integral we obtain

$$\operatorname{Re}\sigma_L(\omega) = \pi\delta(\omega)\partial_\lambda^2 f(\lambda)\big|_{\lambda=0} + \frac{1-e^{-\frac{\omega}{T}}}{2\omega L} \int_{-\infty}^{\infty} dt \; e^{i\omega t} \langle \jmath(t)\jmath \rangle_T$$

$$= \pi\delta(\omega)\partial_\lambda^2 f(\lambda)\big|_{\lambda=0} + \frac{e^{\frac{\omega}{T}}-1}{2\omega L} \int_{-\infty}^{\infty} dt \; e^{-i\omega t} \langle \jmath(t)\jmath \rangle_T. \tag{69}$$

From here it is obvious that $\operatorname{Re}\sigma_L(\omega)$ is even. Since the Meissner fraction vanishes in the thermodynamic limit, we obtain the formula

$$\operatorname{Re}\sigma(\omega) = \lim_{L\to\infty} \operatorname{Re}\sigma_L(\omega) = \frac{1-e^{-\frac{\omega}{T}}}{2\omega} \int_{-\infty}^{\infty} dt \; e^{i\omega t} \lim_{L\to\infty} \frac{\langle \jmath(t)\jmath \rangle_T}{L} \tag{70}$$

that is used in the main text.

## C  Two-spinon contribution

### C.1  Two-spinon dynamical structure function

Defining

$$S_\jmath^{(2\ell)}(Q,\omega) = \sum_{m=-\infty}^{\infty} \int_{-\infty}^{\infty} dt \; e^{i(Qm+\omega t)} C^{(2\ell)}(m,t) \tag{71}$$

the function

$$S_\jmath(Q,\omega) = \sum_{\ell=1}^{\infty} S_\jmath^{(2\ell)}(Q,\omega) \tag{72}$$

is called the dynamical structure function of the local magnetic currents. In the following we shall obtain an explicit expression for the one-particle one-hole term $S_\jmath^{(2)}(0,\omega)$.

For this purpose we start with a close inspection and simplification of the amplitude

$$\mathcal{A}_\jmath^{(2)}(u,v|k) = \left( \frac{(\varepsilon(u)-\varepsilon(v))\vartheta_1'}{4\sin(u-v)\vartheta_1(\Sigma)} \right)^2 \frac{\Xi^2(0)\mathcal{M}\hat{\mathcal{M}}}{\Xi(u-v)\Xi(v-u)}, \tag{73}$$

where

$$\Sigma = -\frac{1}{2}(u-v+\pi k), \quad \Xi(z) = \frac{\Gamma_{q^4}\left(\frac{1}{2}+\frac{z}{2i\gamma}\right) G_{q^4}^2\left(1+\frac{z}{2i\gamma}\right)}{\Gamma_{q^4}\left(1+\frac{z}{2i\gamma}\right) G_{q^4}^2\left(\frac{1}{2}+\frac{z}{2i\gamma}\right)}, \tag{74a}$$

$$\mathcal{M} = \overline{\Phi}_1(P;0) - \Phi_1(P;0)\frac{\phi^{(-)}(v)}{\phi^{(+)}(v)}, \quad \hat{\mathcal{M}} = \hat{\Phi}_1(H;0) - \hat{\overline{\Phi}}_1(H;0)\frac{\phi^{(-)}(u)}{\phi^{(+)}(u)}, \tag{74b}$$

with $H = e^{2iu}, P = e^{2iv}$ and

$$\frac{\phi^{(-)}(v)}{\phi^{(+)}(v)} = \frac{\phi^{(-)}(u)}{\phi^{(+)}(u)} = e^{-2i\Sigma} \frac{\Gamma_{q^4}\left(\frac{1}{2} + \frac{i(v-u)}{2\gamma}\right)\Gamma_{q^4}\left(1 - \frac{i(v-u)}{2\gamma}\right)}{\Gamma_{q^4}\left(\frac{1}{2} - \frac{i(v-u)}{2\gamma}\right)\Gamma_{q^4}\left(1 + \frac{i(v-u)}{2\gamma}\right)}, \tag{75a}$$

$$\Phi_1(P;0) = \hat{\bar{\Phi}}_1(H;0) = {}_2\Phi_1\left(\begin{matrix} q^{-2}, P/H \\ q^2 P/H \end{matrix} ; q^4, q^4\right), \tag{75b}$$

$$\overline{\Phi}_1(P;0) = \hat{\Phi}_1(H;0) = {}_2\Phi_1\left(\begin{matrix} q^{-2}, H/P \\ q^2 H/P \end{matrix} ; q^4, q^4\right). \tag{75c}$$

Using the $q$-Gauß identity [70],

$$_2\Phi_1\left(\begin{matrix} q^{-2}, H/P \\ q^2 H/P \end{matrix} ; q^4, q^4\right) = \frac{\Gamma_{q^4}\left(\frac{1}{2} + \frac{i(v-u)}{2\gamma}\right)}{\Gamma_{q^4}\left(\frac{1}{2}\right)\Gamma_{q^4}\left(1 + \frac{i(v-u)}{2\gamma}\right)}, \tag{76}$$

as well as the functional equations for the $q$-gamma and $q$-Barnes functions, the amplitude can be rewritten as

$$\mathcal{A}_{\partial}^{(2)}(u, v + i\gamma|k) = \left(\frac{\varepsilon(u) + \varepsilon(v)}{2}\right)^2 \frac{(-1)^k q^{\frac{1}{2}} \operatorname{tg}\left(\frac{1}{2}(u - v - i\gamma + \pi k)\right)}{2\sin(u - v)}$$
$$\times B(u - v)\left(\frac{\vartheta_1'}{\vartheta_4\left(\frac{1}{2}(u - v + \pi k)\right)}\right)^2, \tag{77}$$

where $B(z)$ was defined in equation (30) of the main text. Note that $B(z)$ has a double zero at $z = 0$. Hence, the simple pole at $u = v$ stemming from the sine function is canceled by a double zero of $B(u - v)$.

For this reason we can write

$$C^{(2)}(m, t) = \sum_{k=0,1} \int_{-\frac{\pi}{2}}^{\frac{\pi}{2}} \frac{dz_1}{2\pi} \int_{-\frac{\pi}{2}}^{\frac{\pi}{2}} \frac{dz_2}{2\pi} \mathcal{A}_s^{(2)}(z_1, z_2|k) e^{imk\pi + i\sum_{j=1}^2 (t\varepsilon(z_j) - mp(z_j))}, \tag{78}$$

where

$$\mathcal{A}_s^{(2)}(z_1, z_2|k) = \frac{1}{2}\left(\mathcal{A}_{\partial}^{(2)}(z_1, z_2 + i\gamma|k) + \mathcal{A}_{\partial}^{(2)}(z_2, z_1 + i\gamma|k)\right)$$
$$= \frac{q^{\frac{1}{2}}}{2}\left(\frac{\varepsilon(z_1) + \varepsilon(z_2)}{2}\right)^2 \frac{B(z_1 - z_2)}{\Delta + (-1)^k \cos(z_1 - z_2)} \frac{(\vartheta_1')^2}{\vartheta_4^2\left(\frac{1}{2}(z_1 - z_2 + \pi k)\right)}. \tag{79}$$

It follows that

$$S_{\partial}^{(2)}(Q, \omega) = \sum_{k=0,1} \int_{-\frac{\pi}{2}}^{\frac{\pi}{2}} dz_1 \int_{-\frac{\pi}{2}}^{\frac{\pi}{2}} dz_2 \, \mathcal{A}_s^{(2)}(z_1, z_2|k)$$
$$\times \delta_{2\pi}\left(Q - p(z_1) - p(z_2) + \pi k\right)\delta\left(\omega + \varepsilon(z_1) + \varepsilon(z_2)\right), \tag{80}$$

where $\delta_{2\pi}$ is a $2\pi$-periodic delta function.

We now substitute

$$\begin{pmatrix} z_1 \\ z_2 \end{pmatrix} \mapsto \begin{pmatrix} \lambda \\ P \end{pmatrix} = \begin{pmatrix} \frac{1}{2}(p(z_1) - p(z_2)) \\ p(z_1) + p(z_2) \end{pmatrix}. \tag{81}$$

For the substitution recall [35] that $p(x)$ is monotonically increasing on $[-\pi/2, \pi/2]$ with $p(-\pi/2) = 0$, $p(\pi/2) = \pi$. Furthermore, the inverse function can be written as

$$p^{-1}(y) = -\frac{\pi}{2K} \operatorname{arcsn}\big(\cos(y)\big|k\big),\tag{82}$$

where $k$ is the elliptic module and $K$ the complete elliptic integral (see (18)). Setting

$$A(P, \lambda|k) = \frac{\mathcal{A}_s^{(2)}\big(p^{-1}(P/2 + \lambda), p^{-1}(P/2 - \lambda)\big|k\big)}{p'\big(p^{-1}(P/2 + \lambda)\big)p'\big(p^{-1}(P/2 - \lambda)\big)}\tag{83}$$

we obtain

$$S_{\mathcal{J}}^{(2)}(Q, \omega) = \sum_{k=0,1} \left\{ \int_0^\pi dP \int_{-\frac{P}{2}}^{\frac{P}{2}} d\lambda + \int_\pi^{2\pi} dP \int_{-\pi+\frac{P}{2}}^{\pi-\frac{P}{2}} d\lambda \right\} A(P, \lambda|k)$$
$$\times \delta_{2\pi}\big(Q - P + \pi k\big)\delta\big(\omega + \varepsilon(p^{-1}(P/2 + \lambda)) + \varepsilon(p^{-1}(P/2 - \lambda))\big).\tag{84}$$

In the latter equation the $P$ integration is now trivial. For $Q \in (0, \pi)$ we obtain

$$S_{\mathcal{J}}^{(2)}(Q, \omega) = \int_{-\frac{Q}{2}}^{\frac{Q}{2}} d\lambda\, A(Q, \lambda|0)\delta\big(\omega + \varepsilon(p^{-1}(Q/2 + \lambda)) + \varepsilon(p^{-1}(Q/2 - \lambda))\big)$$
$$+ \int_{-\frac{\pi-Q}{2}}^{\frac{\pi-Q}{2}} d\lambda\, A(Q + \pi, \lambda|1)\delta\big(\omega + \varepsilon(p^{-1}(\tfrac{Q+\pi}{2} + \lambda)) + \varepsilon(p^{-1}(\tfrac{Q+\pi}{2} - \lambda))\big).\tag{85}$$

In the limit $Q \to 0$ the first integral can at most contribute to the value of $S_{\mathcal{J}}^{(2)}$ at the single point $(0, -2\varepsilon(-\pi/2))$. We shall ignore this irregular contribution. Taking into account that $A(P, \lambda|k) = A(P, -\lambda|k)$ we see that at all other points

$$S_{\mathcal{J}}^{(2)}(0, \omega) = 2\int_0^{\frac{\pi}{2}} d\lambda\, A(\pi, \lambda|1)\delta\big(\omega + \varepsilon(p^{-1}(\tfrac{\pi}{2} + \lambda)) + \varepsilon(p^{-1}(\tfrac{\pi}{2} - \lambda))\big).\tag{86}$$

Further noticing that

$$\varepsilon(p^{-1}(\tfrac{\pi}{2} \pm \lambda)) = -\frac{h_\ell}{2k'}\sqrt{1 - k^2 \sin^2(\lambda)} \quad \text{and} \quad \varepsilon(\lambda) = -2J \sinh(\gamma)p'(\lambda),\tag{87}$$

see (A.11) of [52] and (A.19) of [35] for more details, we can readily calculate the remaining integral. Using (30) we arrive at

$$S_{\mathcal{J}}^{(2)}(0, \omega) = \frac{q^{\frac{1}{2}}h_\ell^2 k}{4k'}\frac{B\big(r(\omega)\big)}{\Delta - \cos\big(r(\omega)\big)}\frac{\vartheta_3^2}{\vartheta_3^2\big(r(\omega)/2\big)}\frac{\omega}{\sqrt{\big((h_\ell/k')^2 - \omega^2\big)\big(\omega^2 - h_\ell^2\big)}},\tag{88}$$

for $\omega \in [h_\ell, h_\ell/k']$. Outside this interval the function $S_{\mathcal{J}}^{(2)}(0, \omega)$ vanishes.

The first integral on the right hand side of (85) can at most contribute to $S_{\mathcal{J}}^{(2)}(0, \omega)$ at $\omega = -2\varepsilon(p^{-1}(0)) = h_\ell$, which means exactly at the lower band edge.

---

One should incorporate $2\pi$ into $p'$ appearing in these works due to the different conventions we use here for the dressed momentum.

### C.2 Two-spinon optical conductivity

We would like to connect the two-spinon contribution to the structure function with the real part of the optical conductivity. For this purpose we first note that

$$\langle \mathfrak{J}_1(t)\mathfrak{J}_{m+1}\rangle = \left(\langle \mathfrak{J}_1(-t)\mathfrak{J}_{m+1}\rangle\right)^*. \tag{89}$$

In order to see this we start with a finite chain of length $L$ for which

$$\left(\langle \mathfrak{J}_1(-t)\mathfrak{J}_{m+1}\rangle\right)^* = \langle (\mathfrak{J}_1(-t)\mathfrak{J}_{m+1})^\dagger\rangle = \langle \mathfrak{J}_{m+1}\mathfrak{J}_1(-t)\rangle$$
$$= \langle \mathfrak{J}_{m+1}(t)\mathfrak{J}_1\rangle = \langle \mathfrak{J}_L(t)\mathfrak{J}_{L-m}\rangle = \langle \mathfrak{J}_1(t)\mathfrak{J}_{m+1}\rangle. \tag{90}$$

Here we have used the invariance under parity transformations in the last equation. Equation (90) holds for every finite $L$, hence also in the thermodynamic limit.

Now (89) implies

$$\langle \mathfrak{J}_1(t)\mathfrak{J}_{m+1}\rangle = \sum_{\ell=1}^{\infty}\left(C^{(2\ell)}(m,-t)\right)^*. \tag{91}$$

Thus, the two-spinon contribution to the correlation function of the total currents per lattice site is

$$\left(2\sum_{m=0}^{\infty}\langle \mathfrak{J}_1(t)\mathfrak{J}_{m+1}\rangle_T - \langle \mathfrak{J}_1(t)\mathfrak{J}_1\rangle_T\right)^{(2)} = \sum_{m=0}^{\infty}C^{(2)}(m,t) + \sum_{m=1}^{\infty}\left(C^{(2)}(m,-t)\right)^* = \sum_{m=-\infty}^{\infty}C^{(2)}(m,t), \tag{92}$$

as can be seen from taking the complex conjugation of the explicit expression (78). Hence, with (24) and (71),

$$\mathrm{Re}\,\sigma^{(2)}(\omega) = \frac{S_{\mathfrak{J}}^{(2)}(0,\omega)}{2\omega}, \tag{93}$$

which is valid for all $\omega > 0$ and $T = 0$. Lemma 3 and Eq. (31) in the main text therefore follow directly from Eq. (88).

### C.3 The isotropic limit: $\ell = 1$

We consider $\mathrm{Re}\,\sigma^{(2)}(\omega)$ near the lower 2-spinon band edge in the isotropic limit $\gamma \to 0$. A convenient parameter in this limit is $q' = \mathrm{e}^{-\frac{\pi^2}{2\gamma}}$ which approaches zero quickly. The energy gap $\Delta\varepsilon$ in the absence of a magnetic field is

$$\Delta\varepsilon = -\varepsilon(\frac{\pi}{2}) \sim \frac{8J\pi\,\mathrm{sh}\,\gamma}{\gamma}q'.$$

Our main goal is to show that the peak location $\omega^*$ of $\mathrm{Re}\,\sigma^{(2)}$ is parameterized as $\omega^* \sim Cq'$ while the corresponding height is given by $\mathrm{Re}\,\sigma^{(2)}(\omega^*) \sim C'/q'$ for some constants $C, C'$.

The various constants behave in this limit as follows,

$$k \sim 1, \qquad\qquad k' \sim 4q',$$
$$K(k) \sim \frac{\pi^2}{2\gamma}, \qquad\qquad h_\ell \sim \frac{16J\pi\,\mathrm{sh}\,\gamma}{\gamma}q'.$$

The upper 2-spinon band edge $\frac{h_\ell}{k'}$ thus reaches $4J\pi$.

Note that $r(\omega) = \pi$ at the lower 2-spinon band edge, $\omega = h_\ell$. Then we conveniently parameterize

$$r(\omega) = \pi - \frac{\gamma}{\pi}\epsilon_r, \qquad\qquad \omega = h_\ell(1 + \epsilon'). \qquad\qquad (94)$$

We assume that $\epsilon_r, \epsilon'$ are $O(1)$. They are not independent but constrained by (30a),

$$\mathrm{sn}(K + \frac{K}{\pi^2}\gamma\epsilon_r|k) = \mathrm{cd}(\frac{K}{\pi^2}\gamma\epsilon_r|k) = \frac{\sqrt{(\frac{h_\ell}{k'})^2 - \omega^2}}{\frac{h_\ell k}{k'}}.$$

By expanding both sides up to $O((q')^2)$, we find that $\epsilon_r$ and $\epsilon'$ are related by

$$\mathrm{ch}\,\epsilon_r - 1 = 4(\epsilon' + \frac{(\epsilon')^2}{2}).$$

In particular, when both of them are infinitesimally small, $\epsilon_r \sim 2\sqrt{2\epsilon'}$. This essentially explains the $\sqrt{\omega^2 - h_\ell^2}$ behaviour of $\mathrm{Re}\,\sigma^{(2)}(\omega)$ for generic $\gamma$.

Now that $B(\pi + z) = B(z)$, we have in this limit,

$$B(r(\omega)) = B(\frac{\gamma\epsilon_r}{\pi}) \sim \frac{1}{G^4(\frac{1}{2})}\frac{\epsilon_r}{2\pi^2}\,\mathrm{sh}\,\frac{\epsilon_r}{2}\prod_{\sigma=\pm 1}\frac{G^2(1 + \frac{\sigma\epsilon_r}{2i\pi})}{G(\frac{1}{2} + \frac{\sigma\epsilon_r}{2i\pi})G(\frac{3}{2} + \frac{\sigma\epsilon_r}{2i\pi})},$$

where $G$ is the (undeformed) Barnes $G$ function. The limits of the other factors in (31) are easily expressed in terms of $\epsilon_r$,

$$\frac{\vartheta_3^2}{\vartheta_3^2(r(\omega)/2)} \sim \frac{1}{4q'\,\mathrm{ch}^2\,\frac{\epsilon_r}{2}},$$

$$\frac{1}{\sqrt{((h_\ell/k')^2 - \omega^2)(\omega^2 - h_\ell^2)}} \sim \frac{k'}{h_\ell^2 k\,\mathrm{sh}\,\frac{\epsilon_r}{2}},$$

$$\frac{1}{\Delta - \cos(r(\omega))} \sim \frac{1}{2}.$$

All in all, $\mathrm{Re}\,\sigma^{(2)}(\omega)$ behaves near $h_\ell$ in the rational limit as

$$\mathrm{Re}\,\sigma^{(2)}(\omega) \sim \frac{1}{128q'G^4(\frac{1}{2})\pi^2}\mathcal{F}(\epsilon_r), \quad \mathcal{F}(x) = \frac{x}{\mathrm{ch}^2\,\frac{x}{2}}\prod_{\sigma=\pm 1}\frac{G^2(1 + \frac{\sigma x}{2i\pi})}{G(\frac{1}{2} + \frac{\sigma x}{2i\pi})G(\frac{3}{2} + \frac{\sigma x}{2i\pi})}.$$

Numerically, $\mathcal{F}(\epsilon_r)$ has a maximum at $\epsilon_r \sim 1.3508$ and the corresponding peak location is $\omega^* \sim 1.2369 h_\ell$. Therefore we conclude that $\mathrm{Re}\,\sigma^{(2)}(\omega^*)$ behaves as $1/q'$, while $\omega^*$ behaves as $q'$. The above argument only takes into account the contribution from the $\ell = 1$ sector but we expect that the higher excitations do not alter the qualitative behavior in this limit.

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
