# Peer review of "Spin conductivity of the XXZ chain in the antiferromagnetic massive regime"

_SciPost Physics, doi:SciPost Phys. 12, 158 (2022)_

## Round 1 · Referee Report · Anonymous (Referee 1) · 2022-3-2

Strengths

  • exact results for the spin conductivity of an interacting system

Weaknesses

  • "exactness" of the results is not really checked
  • some comments on references and other minor points

Report

The paper presents solid results on calculations that some of the authors are pushing since a number of years. I have only minor comments:

1- the "exactness" of the results is not well checked. Figure 1 does not show a comparison with quasi-exact dmrg calculations to their precision (i.e. 10^-6). In principle there could be deviations of order 10^-1 not visibles from the figure. 2- The values of velocities in table 1 do not seem to make much sense, (they should be not larger than 2) or at least they do not agree with other approaches (i.e. TBA). Maybe the author consider a different rescaling of the Hamiltonian or a value of coupling J not equal to 1? 3- The DC conductivity, i.e. sigma(omega->0^+), is very well know to be zero at T=0 in gapped XXZ. This is not really evident from Fig. 4. Comments? - Comments on the text:

"The theoretical emphasis in recent years was on a systematic justification of phenomenological approaches and on numerical work in a setting that often attempted to go beyond the framework of linear response." This sentence is not clear and possibly misleading. The authors here are considering a linear response quantity (conductivity)

" The calculation of the spin conductivity requires an honest calculation of the dynamical correlation function of two spin-current density operators." It is not clear what is an honest calculation.

"So far, the most successful attempts to exactly calculate dynamical correlation functions of the XXZ chain were based on different types of form factor series expansions." I understand the authors here refer to their attempts. However, it would be preferable to refer also to other "successful attempts" of the past years. I invite the authors to refer to ref [2], in case they are interested only in the most recent review paper, for calculations of spin diffusion constants and Drude weights, at any temperature and magnetic field.

"The Lorentziantype peak around ω = 0 observed in this paper, which seems to decrease with increasing T, therefore appears to be a genuine finite-temperature effect related to the expected diffusive behavior. " This sentece seems to ignore all the theoretical work on the spin diffusion at half filling in gapped XXZ. Again, I invite the authors to consult ref [1] and [2] and references inside. It is indeed well established that at finite T DC conductiviy is finite (and its value is known) , namely sigma(omega) has a Lorentziantype peak. Therefore, this reported conclusion is far from new.

Requested changes

  • Address point 1 and 3 by slightly modifying the figures. Comment on point 2. Reformulate the text when needed.

---

## Round 1 · Referee Report · Anonymous (Referee 2) · 2022-3-4

Strengths

it is a controlled and accurate evaluation of the spin conductivity in the easy axis spin-1/2 Heisenberg chain at T=0

Weaknesses

technically very involved

Report

In this work the authors study the spin conductivity of the Heisenberg chain in the massive regime, extending a thermal form factor approach
they have introduced in ref. 27.
It is actually a sequel of works where they developed and applied this approach, ref. 34+35.
It has the advantage of avoiding the introduction of string states
giving a clear picture of the processes involved.
They further corroborate their analysis with numerical light-cone renormalization group (LCRG) simulations.
They presentation is well structured and clear, nevertheless the analysis
is technically fairly involved and based on previous work by the authors.

Requested changes

It should be clarified in the abstract that it is a T=0 calculation

---

## Round 1 · Referee Report · Anonymous (Referee 3) · 2022-3-23

Strengths

1- Fast converging series expansion for a particular set of correlation functions in quantum integrable models

Weaknesses

1- The paper is not self-contained, as the derivation of the results (and more general ones) will be presented in a forthcoming publication 2- The state of the art regarding spin transport, Drude weights, etc..., in XXZ chains, is very vaguely referred to.

Report

The authors expand on a previous publication, Ref. [27], which introduced a novel approach to the computation of correlation functions in quantum integrable models, based on a summation of form factors in a transverse channel (associated with the "quantum transfer matrix"). Ref. [27] was concerned with correlation functions of operators acting on one lattice site.
A forthcoming technical publication will generalize the results of [27] to more general operators, and the present work borrows from this in order to present results for the dynamical correlations of spin current (in the antiferromagnetic regime, and at zero temperature), which has been the subject of much attention due to its relation with spin transport.

This paper is not really self-contained, as the central formulae (2.11) and (2.12) are presented without any derivation. However, due to its excellent convergence properties, the series representation (2.11) is a very practical tool which allows the authors to draw some physical conclusions, and hence may deserve publication on its own.

Requested changes

1- The abstract should make clear that the results here are for the ground state, or, said differently, at zero temperature

2-The introduction should describe in more detail the relation between the Drude weight, conductivity and current dynamical correlation functions, paying some attention to discussing which conclusions hold for generic non-zero temperatures, and which hold at $T=0$.

3- The above point also applies to the discussion around Fig. 2 : according to the authors the observations lade on the current autocorrelation indicate a non-ballistic transport. It should be recalled what to expect in the case of ballistic transport (plateau at large times ? any further subtleties at zero temperature ?)

4- On p.6 there seems to be two typos in the beginning of the second paragraph : $N_0$ should be $N_{>0}$, and "large $m$" should be "large $L_c$"

5- eqs. (3.1) and (3.2) are singular at $T=0$, so the authors should clarify how to use these formulae at $T=0$.

6- This paper has the potential to interest readers not too familiar with integrability, but mostly interested in the physical conclusions and transport properties. For those readers, it would be instructive, and make the paper a little more self-contained, if a short recap on the "thermal form factor" approach could be given, and an explanation why it works so well with respect to other methods. One may wonder if the quality of the series convergence has to do with the "transverse channel" used in this method, just like in quantum transfer matrix methods the thermodynamic limit projects onto the leading eigenstate. Is this indeed the case ? If so, I think this would be worth making explicit.

---

## Round 2 · Referee Report · Anonymous (Referee 3) · 2022-4-14

Report

I thank the authors for the changes brought to the manuscript, I now recommend submission in the present form.

---

## Round 2 · Referee Report · Anonymous (Referee 1) · 2022-4-14

Report

The paper is now ready to be published.

---

## Round 2 · Referee Report · Anonymous (Referee 2) · 2022-4-15

Report

the paper can now be published as is

---

## Round 2 · Author Response

Following the editor's recommendation the manuscript underwent
a major revision. The main change concerns the introduction which
is now considerably enlarged such as to allow for the citation of more
background material. In addition, we have addressed all points that
were explicitly raised by the referees and some points that were implicit
in their statements.

Here are our comments to the points raised by the referees:

Referee 1:

Minor comments:

1 - No exact error estimates can be obtained at long times in time-dependent DMRG calculations. As a substitute, one uses, on the one hand, the truncation error - given by the sum of the eigenvalues belonging to the eigenstates of the reduced density matrix which are truncated in a single renormalization group step - and, on the other hand, a comparison of simulations where a different number of states have been kept. The truncation error and the number of states kept are given in the manuscript. For the longest simulation times, we do indeed only claim an estimated accuracy of the order of the symbol sizes in the figure which are ~10^{-1}. This is spelled out explicitly as well. We believe that our results for the achieved simulation times and accuracies are in-line with the best available tDMRG codes. tDMRG calculations at long times are not quasi-exact and it is not possible to estimate the error to a precision of 10^{-6}.

2 - Thank you for raising this point. The relation with Section 2.3
was not clearly explained. The velocity v was actually taken as
v_2 from Section 3.2. We have revised the text at the bottom
of page 7 and the caption of the figure showing Table 1, where we
now refer to Eq. (2.17).

3 - The analytical result for the two-spinon contribution is indeed zero for omega->0. The purpose of Fig. 4 was to show that the obtained results for the time-dependent current-current correlation function are so precise that even a direct Fourier transform without any filters or windowing functions yields only very small deviations from zero at zero frequency. This has been further clarified in the amended manuscript.

Comments on the text:

" ... attempted to go beyond the framework of linear response"
-> We have removed this sentence as we understand it can be
mistakable here.

" ... requires and honest calculation of ..."
-> This sentence has been removed as well as we understand that
it may appear slightly polemic to some readers.

"So far, the most successful attempts ..."
-> This sentence refers to exact results at finite frequency for
the XXZ chain. Here we have referred to everything relevant of that
kind we were aware of. Most of the articles cited here are not
our attempts but belong to other authors. We think it would be
misleading to list works for zero frequency here.

We understand, however, that it might be helpful for the readers
to place our work into the context of those works as was also
suggested by the third referee.

In order to comply with this suggestion we have split the
paragraph following Eq. (1.1) and have considerably extended the
discussion of spin transport. Now the ballistic component and
the Drude weight are discussed and reference to the corresponding
literature is made (see bottom part of page 2 and top part of
page 3).

"The Lorentzian peak around ..."

We are not aware of any previous exact results for the spin conductivity at finite frequencies.

Referee 2:

We have clarified in the abstract that we are discussing the
limit of zero temperature. Thank you for this suggestion.

Referee 3:

1 - done (see reply to second referee)
2 - done (see reply to first referee)
3 - we have changed the text accordingly and hope that this
is clearer now
4 - corrected, thank you!
5 - We do not think that these formula are singular, but the
case omega = 0 would require a separate consideration. In
Appendix B we show that the function sigma' is even in omega.
For this reason we concentrate on omega > 0, which in the
revised version is stated in several place in the text e.g. on
pages 10 and 11.
6 - We think that the massive nature of the excitations is
responsible for the fast convergence at T = 0. The true
advantages of the method will become apparent at T > 0, where
there is indeed a sum less, which is replaced by a projection
onto a single dominant state, just like in case of the thermodynamics
and of the static correlation functions. Amazingly, in the
limit T -> 0 another advantage appears. It does not affect
the relative size of the terms in the series but the structure
of the integrands in the multiple integrals. In our understanding
this is an important technical achievement on the way
from [46] (which was [27] in the previous version) to [53,54]:
As compared to previous methods the integrand is explicit in every
term and no additional integrals or sums appear in the definitions
of the integrands. It is this feature which makes the integrals
computable and gives us hope that we can prove the convergence
of the series and derive explicit error estimates. We have tried
to make this more clear by extending the second paragraph on
page 3 and the Summary section.

We have also included a number of remarks on how to derive our main
formulae (2.11), (2.12) in the last paragraph of the introduction.
Some experts may now understand this point. We hope that this
may easy, at least partially, the dissatisfaction uttered by the referee,
when he/she says that "this paper is not really self-contained". In
fact, the derivation of the main formulae will be presented elsewhere,
but the methods required to derive them were fully laid out in our
previous work.

---

## Round 2 · List of Changes

1) "at zero temperature" appended to the first sentence of the abstract.

2) We replaced the term "dynamical conductivity" by "optical conductivity" as this seems more common in the physics literature.

3) Second sentence of the first paragraph of the introduction "The theoretical emphasis ..." deleted and first and second paragraph merged.

4) Former third paragraph split into a part on the thermal Drude weight and another one on the spin Drude weight.

5) Text starting from "Still, it may have..." on the bottom of page 2 of the revised manuscript to "... does not involve any kind of string hypothesis" in the middle of page 3 is new. In this new part of the text we now discuss ballistic and diffusive spin transport at T > 0 and place our work into the context of the cited results.

6) Phrase "... and can be interpreted as a resummation of the latter" added in the third paragraph on page 3.

7) Sentences from "This has to be contrasted with..." to "... massive nature of the excitations" at the end of the third paragraph on page 3 added. We added this in order to comply with the third referee's suggestion to explain more details of the method.

8) For the same reason we added "Although the general formula is easy to guess ..." and the sentences from "For this special case the result may be obtained ..." to the first paragraph on page 4.

9) On page 6 we added the second paragraph of section 2.2 starting with "In order to estimate the truncation error ...". We did this to clarify the question about the numerical error raised by the first referee.

10) Typo in the second line of third paragraph of section 2.2 corrected.

11) End of the third paragraph of section 2.2 starting from "... in agreement with common belief ..." modified and amended.

12) Last paragraph of section 2.2 and Table 1 modified and amended "$v$" changed into "$v_2$" reference to section 2.3 added.

13) At two places on page 11 "$\omega > 0$" added.

14) Text of the second paragraph below equation (3.8) starting with "Note that we perform ..." amended.

15) Last sentence "We attribute the fast convergence ..." added to the first paragraph of section 4.

16) Last paragraph starting with "Further future goals ..." added to section 4.

17) References [17-35] and [69] added.

18) In addition some minor typos corrected and graphical improvements performed.

---

## Editorial Decision

published